# Determination of the Standard Visual Criterion for Diagnosing and Treating Presbyopia According to Subjective Patient Symptoms

**DOI:** 10.3390/jcm10173942

**Published:** 2021-08-31

**Authors:** Yukari Tsuneyoshi, Sachiko Masui, Hiroyuki Arai, Ikuko Toda, Miyuki Kubota, Shunsuke Kubota, Kazuo Tsubota, Masahiko Ayaki, Kazuno Negishi

**Affiliations:** 1Department of Ophthalmology, Keio University School of Medicine, Tokyo 160-8582, Japan; yukari.a7@keio.jp (Y.T.); m.sac@a7.keio.jp (S.M.); tsubota@z3.keio.jp (K.T.); 2Queen’s Eye Clinic, Yokohama 220-6204, Japan; arai@minatomiraieye.jp; 3Minamiaoyama Eye Clinic, Tokyo 107-0061, Japan; toda@minamiaoyama.or.jp; 4Department of Ophthalmology, Shonan Keiiku Hospital, Fujisawa 252-0816, Japan; myu.kubota@gmail.com (M.K.); shun_kubota@live.jp (S.K.); 5Otake Clinic Moon View Eye Center, Yamato 242-0001, Japan

**Keywords:** presbyopia, near visual acuity, standard criterion, diagnosis

## Abstract

Presbyopia treatments using various modalities have been developed recently; however, no standard criteria exist for the diagnosis and treatment endpoint. This study assessed the relationship between the near visual acuity (NVA) and the subjective symptoms of phakic presbyopia and determined the numerical NVA threshold to diagnose phakic presbyopia and evaluate the effectiveness of presbyopia treatment. The binocular distance, NVA with habitual correction, and monocular conventional VA were measured. Patients were asked about their awareness of presbyopia and difficulty performing near tasks. This prospective observational study included 70 patients (mean age, 56 years; range, 32–77). Most patients became aware of presbyopia in their late forties, although some had difficulty with vision-related near tasks before becoming aware of presbyopia. Eighty three percent of patients (20/24) experienced difficulty with near vision-related tasks even with excellent NVA at 40 cm with habitual correction of 0.0 logMAR (20/20 in Snellen VA). In conclusion, the current study showed that patients became aware of presbyopia in their late forties, although some had difficulty with near vision-related tasks before becoming aware of presbyopia. Further investigation should include the proposal of appropriate diagnostic criteria for presbyopia and better management for patients with presbyopia.

## 1. Introduction

Presbyopia is a global problem that affects about one quarter of the world’s population [1]. The number of people with impaired near vision due to presbyopia is estimated to decrease by about 20% by 2050 because of the increasing myopia prevalence [1]. However, presbyopia remains an important health problem that may affect the quality of life of individuals not only in developing countries without awareness of presbyopia or accessibility to affordable treatment [1,2] but also in developed countries where people tend to be engaged in near tasks because of the increasing use of digital technology.

Several definitions of presbyopia have been used historically, and most were functional or qualitative [1,2]. Previous studies of presbyopia treatment using various treatment modalities [3] used arbitrary numerical criteria to determine treatment efficacy because of the absence of standardized criteria. The patient-reported outcome measures have also been used in clinical trials and quality-of-life studies for presbyopia treatment [3,4]. Among them, the Near Activity Visual Questionnaire was identified as the most appropriate for assessing near-vision functioning in presbyopia, although the measure was not validated in a purely phakic presbyopia sample [4]. Considering recent developments of presbyopia treatments using various modalities [3,5,6,7,8,9], simple and easily accessible standardized criteria for diagnosis and endpoints of treatment are necessary.

The purpose of this study was to assess the relationship between the near visual acuity (NVA) and the subjective symptoms of phakic presbyopia and to determine the numerical NVA threshold to diagnose phakic presbyopia and evaluate the effectiveness of presbyopia correction according to subjective patient symptoms.

## 2. Materials and Methods

### 2.1. Study Design and Patients

This study was a clinic-based prospective observational study conducted at four eye clinics: Minamiaoyama Eye Clinic (Tokyo, Japan), Queen’s Eye Clinic (Kanagawa, Japan), Shonan Keiiku Hospital (Kanagawa, Japan), and Keio University Hospital (Tokyo, Japan). All patients provided written informed consent before participating in this study. The institutional review boards of each institution approved the study (approval numbers, 20181025-2, Minamiaoyama Eye Clinic; 20181025-2, Queen’s Eye Clinic; 18-002, Shonan Keiiku Hospital; and 20150280, Keio University Hospital), which followed the tenets of the Declaration of Helsinki. The protocol for this study was registered with the University Hospital Medical Information Network Clinical Trial Registry (UMIN000021587).

### 2.2. Inclusion and Exclusion Criteria

The inclusion criteria were age 20 years and older, phakia because of the need to measure the refraction and undergo VA tests for diagnosis or treatments, and a binocular distance visual acuity (DVA) of 0.10 logMAR (16/20 in Snellen acuity) and over. The exclusion criteria were a history of refractive surgery and decreased cognitive function.

### 2.3. Ophthalmic Examinations and Questionnaire

Experienced examiners performed all examinations. The ophthalmologic evaluation of the participants included measurement of the monocular corrected DVA (CDVA) and monocular distance-corrected NVA (DCNVA), binocular DVA with habitual correction (DVAHC), and binocular NVA at 40 cm with habitual correction (NVAHC). If a patient did not use any corrective lens for near visual tasks, the binocular NVA was measured without correction. All distance and near VA charts followed the Japanese industrial standards (JIS) T7309 (http://kikakurui.com/t7/T7309-2002-01.html, accessed on 29 August 2021), which is based on International Organization for Standardization (ISO) 8596: 1994, Ophthalmic optics: Visual acuity testing—Standard optotype and its presentation; and ISO 8597: 1994, Optics and optical instruments: Visual acuity testing—Method of correlating optotypes. It was reported that the VA charts that adhere to the JIS are consistent with the ones that adhere to the international standard (http://kikakurui.com/t7/T7309-2002-01.html, accessed on 29 August 2021). When the VA was measured using a decimal VA chart, the measured decimal VA was converted to logarithm of the minimum angle of resolution (logMAR) units according to the VA conversion chart [10]. Using an interview sheet, patients were asked to determine the presence or absence of presbyopic symptoms and the age at which they heard or realized by themselves for the first time that the symptoms they had had for a while represented presbyopia. Patients also were asked about the degree of difficulty while reading a newspaper and reading a book for an extended time. The degrees of difficulty were divided into no difficulty, slight difficulty, and great difficulty.

### 2.4. Statistical Analysis

All statistical analyses were conducted using commercially available statistical software (IBM SPSS Statistics, version 25, Armonk, NY, USA). The Mann-Whitney U test was used to compare the VA and subjective refraction when the data were not normally distributed. The χ^2^ test was used to compare the proportions of patients who were male and female. All tests of statistical significance were two-sided, and *p* < 0.05 was considered statistically significant.

## 3. Results

### 3.1. Patient Profile

The study included 70 patients (30 male and 40 female patients). The mean age was 56.0 ± 13.0 (standard deviation: SD) years old (range: 32–77). The mean monocular subjective refraction (spherical equivalent) of all eyes was −2.78 ± 3.70 (SD) dioptors, and the mean corrected distance visual acuity (CDVA) and DCNVA (distance corrected near visual acuity) at 40 cm were −0.09 ± 0.09 (SD) and 0.28 ± 0.33 (SD), respectively for all 140 eyes of 70 patients. The binocular distance visual acuity with habitual correction (DVAHC) and the near visual acuity with habitual correction (NVAHC) were −0.03 ± 0.12 (SD) and 0.05 ± 0.16 (SD), respectively. In our data, the monocular CDVA of all patients was 0.10 logMAR (16/20 Snellen acuity) and better except for one patient whose CDVA was 0.22 logMAR (12/20) in the right eye and 0.15 logMAR (14/20) in the left eye. However, the binocular DVA with habitual correction (DVAHC) of this patient was 0.2 logMAR (12/20), which was relatively good without being affected by the reduced CDVA.

Figure 1 is a histogram of the patients’ ages.

In some cases, multiple ocular complications developed that included cataract (22 cases), dry eye (20 cases), vitreous macular traction (13 cases), chorioretinal atrophy (13 cases), optic disc cupping (12 cases), age-related macular degeneration (7 cases), conjunctivitis (6 cases), epiretinal membrane (3 cases), and corneal opacity, keratoconus, pterygium, uveitis (1 case each). Those patients with multiple complications developed were asthma and hypertension (3 cases each), hay fever (2 cases), and allergic sinusitis, atopic dermatitis, breast cancer, chronic nephritis, diabetes mellites, endometriosis, fatty liver, heart disease, hypothyroidism, Parkinson’s disease, rheumatic disease, and sarcoidosis (1 case each). No patient had complications that severely affected visual function, meaning 0.40 logMAR (20/40 in Snellen acuity) and worse in CDVA.

### 3.2. Questionnaire Results

The questionnaire showed that 65.7% (46/70) of patients replied that they had presbyopic symptoms. The percentages of patients who were aware of presbyopia by age are shown in Figure 2.

No one was aware of presbyopia before reaching 45 years of age, and the percentages of patients with subjective presbyopia increased dramatically over 45 years of age and plateaued after 55 years. The mean initial age of the patients with subjective presbyopia was 50.9 (standard deviation 7.1; range, 38–70) years. Figure 3 shows the relationship between the level of difficulty when performing near tasks and the percentages of patients 45 years old and over with subjective presbyopia.

The percentage of patients who were aware of presbyopia increased when they considered it very difficult to read a newspaper or book for an extended period compared with those who described no or slight difficulty. However, around 25% of the patients were unaware of presbyopia despite having slight difficulty performing near tasks.

### 3.3. Differences in Subjective Refraction and VAs between Patients with/without Awareness of Presbyopia

Table 1 shows the differences in the subjective refraction and VAs between patients with and without an awareness of presbyopia.

The subjective refraction of the patients who were unaware of presbyopia was significantly more myopic than those who were unaware of presbyopia. Naturally, the binocular NVA with habitual correction was significantly worse in patients who were aware of presbyopia, although there was no significant difference in the binocular DVA with habitual correction.

### 3.4. Relationship between Binocular NVA and Subjective Symptoms

Figure 4 shows the percentages of patients who were aware of presbyopia, those who had difficulty reading a newspaper, and those who had difficulty reading a book for an extended period based on the NVA with habitual correction. 

All of the percentages of patients who were aware of presbyopia and who had difficulty reading a newspaper and difficulty reading a book for an extended period increased dramatically when the binocular NVDAC at 40 cm decreased to 0.0 (20/20).

## 4. Discussion

Several methods can evaluate the visual function in presbyopia including measurement of the NVA, defocus curves, accommodative amplitude, and reading speed. Among them, the NVA test at 40 cm is the most common clinical examination, although the test distances are not standardized.

Holden defined functional presbyopia as the need for addition of a significant optical correction to the presenting distance refractive correction to achieve a NVA absolute (such as N8 or J1) or relative (such as 1 line of acuity improvement) criteria [11]. Other epidemiologic studies have reported that presbyopia is the inability of individuals aged 35 years or older to read binocularly N8 (or 6/12) at 40 cm or their habitual working distance; in some studies, presbyopia was limited to those patients whose NVA improved with the addition of corrective lenses [12,13,14,15]. The Japanese Society of Presbyopia determined two different criteria for diagnosing presbyopia, i.e., medical presbyopia and clinical presbyopia [16]. According to the definition, medical presbyopia is an ocular condition with an accommodative amplitude less than 2.5 diopters (D) regardless of presbyopic symptoms, and clinical presbyopia is an ocular condition in which the NVA is less than 20/50 with habitual correction in addition to the presence of presbyopic symptoms.

Regarding the endpoints for presbyopia treatments, the number of primary eyes with a DCNVA at 40 cm achieving 20/40 or better and a gain of at least 10 letters was adopted in clinical trials of the treatment modality for presbyopia [17,18,19].

According to our results, the percentages of patients who had difficulty with near tasks (reading a newspaper and reading a book for an extended period) and the rate of awareness of presbyopia dramatically increased when the binocular NVA decreased to 20/20. In addition, 83% (20/24) of patients reported difficulty with near vision-related tasks even with an excellent logMAR NVAHC of 0.0 (20/20 in Snellen VA). This indicates that a NVA over 20/20 is necessary for comfortable near vision, and the most common threshold for presbyopia treatment to read binocularly N8 (or 20/40) at 40 cm might be too low as an endpoint of presbyopia correction. Other aspects of vision, such as insufficient accommodation amplitude (for example, less than 4 D), contrast sensitivity, and/or stability of the ocular surface, also may play an important role in determining difficulty performing near visual tasks, and we should pay more attention more to these factors to predict patients’ need of intervention for presbyopia.

It is difficult to diagnose degraded near visual function in subjects with early presbyopia with good conventional VA clinically and quantitatively. We reported previously that the near functional VA (FVA) test detected early presbyopia better than the conventional VA test [20]. The FVA Measurement System, which is commercially available, can calculate the mean VA over 60 s from the VA data measured continuously for 60 s. The near FVA was negatively and significantly linearly correlated with the accommodative amplitude, and the decrease in the near FVA for a reduction of 1.00 D of accommodative power was greater than that of the DCNVA, which means that the near FVA may be a good option for diagnosing presbyopia [20].

Our results showed that the subjective refraction of patients unaware of presbyopia was significantly more myopic than those aware of the presbyopia. Myopic eyes need less accommodative efforts due to the difference between the ocular and spectacle accommodation [21]. Moreover, myopic subjects can use the effect of the forward spectacle shift [21]. These advantages for near vision in myopic eyes might affect the awareness of presbyopia.

Our study also showed that there were many presbyopic patients who were unaware of the presbyopia. Among patients aged 45 years and older, between 15% and 20% were unaware of presbyopia despite difficulty performing near tasks.

Uncorrected presbyopia resulted in significant decreases in productivity and quality of life in the poorest communities [3,12,13,15,22,23,24,25]. Even in developed countries, presbyopia may cause severe health problems due to eye strain and asthenopia [26,27] because of the dramatically increasing use of digital devices. The results of our study implied the presence of considerable uncorrected presbyopia in developed countries, where it is easy to access treatment, due to the lack of awareness of presbyopia. We should enlighten patients regarding the onset, symptoms, effects, and corrective methods to minimize its impact. Establishing a universal, precise diagnostic criteria for presbyopia can result in an appropriate understanding of the burden of presbyopia and need for correction.

The current study had some limitations. First, this study included a small number of cases and, second, we measured only the conventional distance and NVAs. Several important parameters, such as the near point distance, the deviation of the habitual refraction from the best subjective refraction, and the habitual reading distance, were not recorded.

This warrants further investigation that considers potentially relevant factors such as age and pupillary size in more cases with other detailed visual function tests such as the FVA test. However, the conventional near visual test is one of the most common and easily accessible tests to evaluate near visual function. In addition, the use of the NVA with habitual correction is the strength of the current study to investigate the relationship between visual function and subjective symptoms compared with an evaluation using the arbitrary standard correction. Therefore, we believe our results are useful to determine the threshold visual function to diagnose presbyopia, especially in developed countries.

In conclusion, the current study showed that patients became aware of presbyopia in their late forties, although some had difficulty with near vision-related tasks before becoming aware of presbyopia. Surprisingly, 83% (20/24) of patients experienced difficulty with near vision-related tasks even with an excellent logMAR NVAHC of 0.0 (20/20 in Snellen VA). This means that a visual acuity of 20/20 at near distances is not correlated with the level of comfort when performing near tasks for an extended time. This is probably due to accommodative fatigue that occurs more often as accommodative amplitude decreases. Considering the current results, the vision threshold for intervening in presbyopia may have to be set at a stricter level than the present one and we may have to reconsider the goal of the treatment of presbyopia much more than the current criteria, at least in developed countries.

## Figures and Tables

**Figure 1 jcm-10-03942-f001:**
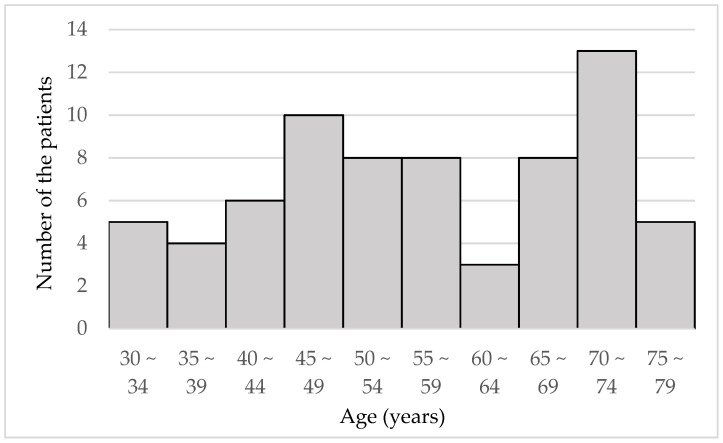
Distribution of patient age (*N* = 70).

**Figure 2 jcm-10-03942-f002:**
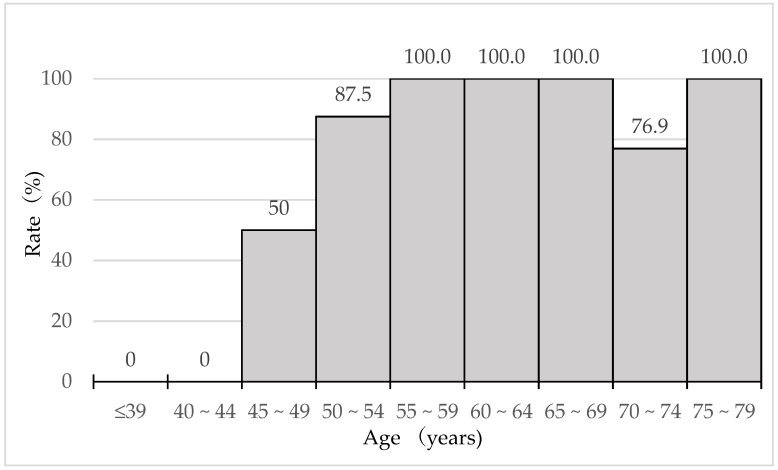
Rate of the patients with awareness of presbyopia.

**Figure 3 jcm-10-03942-f003:**
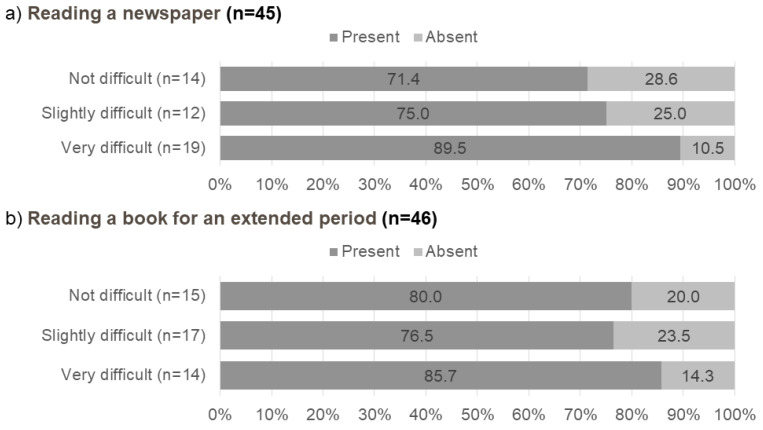
Awareness of presbyopia and subjective symptoms for near tasks in patients aged 45 years and older (*n* = 55).

**Figure 4 jcm-10-03942-f004:**
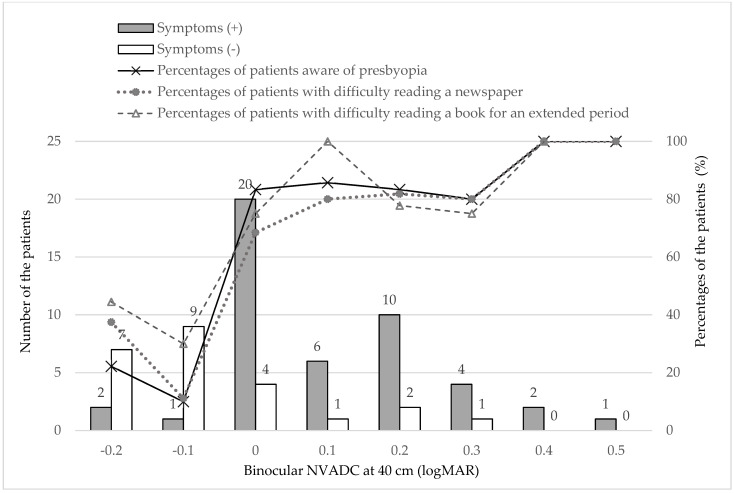
Binocular NVAHC and the rates of the patients with awareness of presbyopia, difficulty in checking a newspaper, and difficulty in reading a book for a long time (*N* = 70).

**Table 1 jcm-10-03942-t001:** Comparisons of the clinical data between the awareness and absence of awareness of presbyopia.

	Aware (*n* = 46)	Unaware (*n* = 24)	*p* Value
Age (years)	62.2 ± 9.7	44.2 ± 12.0	0.000
Sex (male/female)	16/30	14/10	0.059
Monocular examination			
Subjective refraction (SE) of the relatively hyperopic eye (D)	−1.58 ± 3.48	−4.13 ± 3.37	0.005
Subjective refraction (SE) of the relatively myopic eye (D)	−2.21 ± 3.80	−4.80 ± 3.11	0.006
CDVA (logMAR) of the better eye	−0.10 ± 0.08	−0.12 ± 0.08	0.118
CDVA (logMAR) of the worse eye	−0.06 ± 0.09	−0.11 ± 0.10	0.013
DCNVA (logMAR) at 40 cm of the better eye	0.37 ± 0.22	−0.03 ± 0.23	0.000
DCNVA (logMAR) at 40 cm of the worse eye	0.49 ± 0.27	−0.00 ± 0.25	0.000
Binocular examination			
DVAHC (logMAR)	−0.28 ± 0.11	−0.04 ± 0.15	0.434
Binocular NVAHC at 40 cm (logMAR)	0.11 ± 0.15	−0.05 ± 0.13	0.000

D—diopters; CDVA—corrected distance visual acuity; logMAR—logarithm of the minimum angle of resolution; DCNVA—distance corrected near visual acuity; DVAHC—distance visual acuity with habitual correction; NVAHC—near visual acuity with habitual correction.

## Data Availability

Supporting data are not available because consent for sharing data was not obtained.

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
