# Peer review of "Determination of the Standard Visual Criterion for Diagnosing and Treating Presbyopia According to Subjective Patient Symptoms"

_jcm, 2021, doi:10.3390/jcm10173942_

Round 1

Reviewer 1 Report

See attached review comments.

Reviewer 2 Report

In their manuscript titled "Determination of the standard visual criterion for diagnosing and treating presbyopia according to subjective patient symptoms", Tsuneyoshi et al. report on a study which assessed  different parameters of subjective experience in near vision.

On one hand, there are several interesting aspects to this study, which are worth being reported. On the other hand, several important parameters, such as the near point distance, the deviation of the habitual refraction from the best subjective refraction, and the habitual reading distance have apparently not been recorded. The study therefore missed the opportunity to shed further light on the factors contributing the the reported findings.

Some specific points.

** Acuity in healthy eyes is normally significantly better than 20/20. It is therefore no surprise that 20/20 is perceived as a slight visual impairment. The authors may want to discuss this. I suggest to delete the word "surprising" from the abstract.

** Line 65, "age 20 years and older" - if this was the inclusion criterion, why were all participants at least 32 years old?

** Line 66 – Does "bilateral" mean "in each eye separately" (as opposed to "binocular")?

** Line 73 (and other places in the manuscript), "daily correction", what does this mean exactly? Is it their habitual refraction (i.e. the refraction of  the spectacles that they are usually wearing)?

** Line 86, "age at which they first became aware of presbyopia", what does this mean exactly? It could for instance mean "the age at which they experienced presbyopia symptoms for the first time without necessarily knowing that these symptoms are evidence of presbyopia", or "the age at which they heard for the first time that the symptoms they had already had since a while represent presbyopia"

** Line 116, "No patient had complications that severely affected visual function."— How did the authors define "severely", and how did they test for an effect on visual function?

** Table 1 — Given the inhomogeneity of the group of participants (large age range, various levels of ametropia, etc.), I am wondering about the meaningfulness of the values in the table. Could the table be omitted without loosing important information?

** Given that some participants had reduced CDVA, how did the authors disentangle a general reduction of acuity from the effect of presbyopia on near vision acuity?

** Table 2 — Could these data be presented graphically for easier comprehension?

Round 2

Reviewer 1 Report

Consider a different emphasis to the paper or a future paper. It doesn't seem too surprising that near VA is insensitive to the need for a reading add. Perhaps other measures (for example accommodative amplitude of less than 4D) may be much more predictive of patients in need of intervention based on subjective criteria.

Reviewer 2 Report

The authors have satisfactorily taken care of the issues that I had raised in the initial review.

Author Response

Thank you for your comments. I appreciate your efforts for the review.